# Efficacy and Safety of a Tri-Valent Ready-to-Use Porcine Circovirus Type 2a, *Mycoplasma hyopneumoniae* and *Lawsonia intracellularis* Vaccine in Weaned Pigs

**DOI:** 10.3390/vaccines13070681

**Published:** 2025-06-25

**Authors:** Michelle Allen, Frank Roerink, Abigail Crowley, Susan Knetter, Chandra Morgan, Huiling Wei, Ruud Segers

**Affiliations:** 1Intervet Inc. (d/b/a Merck Animal Health), De Soto, KS 66018, USA; abigail.lane@merck.com (A.C.); susan.knetter@merck.com (S.K.); chandra.morgan@merck.com (C.M.); huiling.wei@merck.com (H.W.); 2Intervet International B.V., 5831 AN Boxmeer, The Netherlands; ruud.segers@merck.com

**Keywords:** *Mycoplasma hyopneumoniae*, *Lawsonia intracellularis*, porcine circovirus, combination vaccine

## Abstract

**Background/Objectives**: This study describes multiple trials demonstrating the safety and efficacy of a tri-valent vaccine against diseases caused by Porcine Circovirus Types 2a and 2d (PCV2a, PCV2d), *Mycoplasma hyopneumoniae*, and *Lawsonia intracellularis*. **Methods**: For each of the PCV2a and PCV2d onset of immunity (OOIs) and duration of immunity (DOI) studies, 25 pigs were vaccinated with a tri-valent vaccine and 25 with placebo. After dual challenge with PCV2a and porcine reproductive and respiratory virus (PRRSV) (OOI) or single challenge with PCV2d (OOI and DOI), respectively, viremia and lymphoid depletion data were collected. For each of the *M. hyopneumoniae* OOI and DOI studies, 35 to 70 pigs were vaccinated with the tri-valent vaccine and 35 to 70 with placebo. After *M. hyopneumoniae* challenge, the lungs were scored for disease. For the *L. intracellularis* OOI study, 40 to 50 pigs were vaccinated with the tri-valent and 40 to 50 with placebo. After *L. intracellularis* challenge, the intestines were scored for disease. All pigs were vaccinated at approximately 3 weeks of age, and all placebo vaccines were product matched. **Results**: Vaccinating pigs with a tri-valent vaccine reduced viremia and lymphoid depletion due to PCV2a and PCV2d, reduced lung lesions due to *M. hyopneumoniae* and reduced ileum and colonization scores due to *L. intracellularis*. **Conclusions**: The trials reported here demonstrated the safety and efficacy of the first ready-to-use PCV2, *M. hyopneumoniae*, and *L. intracellularis* vaccine for pigs.

## 1. Introduction

Porcine Circovirus Type 2 (PCV2), *Mycoplasma hyopneumoniae*, and *Lawsonia intracellularis* are three of the most economically significant pathogens in the swine industry worldwide. PCV2 is the causative agent of a number of diseases and syndromes collectively referred to as porcine circovirus-associated disease, with postweaning multisystemic wasting syndrome and porcine respiratory disease complex (PRDC) being the most economically important [1]. Additional disorders associated with PCV2 include reproductive failure, granulomatous enteritis, congenital tremors and exudative epidermitis, with subclinical infections manifesting as poor growth performance [2]. Currently, the most prevalent genotype of PCV2 worldwide has switched from PCV2a to PCV2d [3].

*M. hyopneumoniae* is also one of the main causative pathogens of PRDC and contributes to financial losses in the pig industry due to decreased performance and increased mortality caused by secondary infections [4]. Alone, it is a respiratory pathogen and the main causative agent of enzootic pneumoniae, which is characterized by a chronic, non-productive cough, reduced weight gain and decreased feed conversion efficiency [5].

*L. intracellularis* multiplies in gut enterocytes and is the cause of significant economic losses worldwide by reducing feed conversion and average daily weight gain [6,7]. The disease, porcine proliferative enteropathy (PPE), is characterized by diarrhea, which can lead to an acute hemorrhagic form in young adult pigs and may lead to death. Subclinical infection manifests as reduced weight gain without clear clinical signs. PPE is recognized as multiple forms of disease including porcine intestinal adenomatosis, necrotic enteritis, regional ileitis, and proliferative hemorrhagic enteropathy correlating to chronic disease and disease due to secondary infections [8].

The objective of this study, comprising seven challenge trials, was to evaluate the safety and efficacy of a tri-valent vaccine against diseases caused by PCV2a, PCV2d, *M. hyopneumoniae*, and *L. intracellularis* under laboratory conditions.

## 2. Materials and Methods

### 2.1. Vaccines

Vaccines (CIRCUMVENT^®^ CML, Intervet Inc., Rahway, NJ, USA) containing the baculovirus-expressed open reading frame 2 (*ORF2*) capsid protein of PCV2, inactivated *M. hyopneumoniae*, inactivated *L. intracellularis*, and XSOLVE adjuvant were evaluated for their safety and efficacy. XSOLVE, known in the United States as MICROSOL DILUVAC FORTE, is an oil-in-water emulsion containing mineral oil and Vitamin E. The vaccines were given intramuscularly in a single, 2 mL dose to pigs three weeks of age or older according to the product label. Placebo vaccines contained all of the reagents and antigens of the vaccine except for the antigen fraction being tested. The PCV2 fraction is Baculovirus/PCV recombinant that expresses the *ORF2* gene, which was from a PCV2a parent strain isolated from infected lung tissue by the University of Nebraska Veterinary Diagnostic Laboratory. The *M. hyopneumoniae* Strain 11 fraction was sourced from the Iowa State University Veterinary Diagnostic Laboratory. The *Lawsonia intracellularis* isolate SPAH08 fraction was isolated from infected intestinal tissue from a pig located in Minnesota, USA.

### 2.2. Efficacy Studies

Onset of immunity (OOI) and duration of immunity (DOI) for each of the three antigen fractions were examined in experimental challenge studies (Table 1). Three-week-old pigs from herds seronegative to the challenge organism and free of *M. hyopneumoniae*, *L. intracellularis*, or PCV2, were used for the OOI and DOI studies, with the exception that caesarean-derived, colostrum-deprived pigs were used for the PCV2d DOI study. For each of the OOI and DOI studies, pigs were randomly divided into two groups (vaccine and placebo) prior to vaccination. All pigs were vaccinated with a single, 2 mL dose of either vaccine or placebo at approximately three weeks of age, following weaning. During each animal trial, all pigs were observed daily for general health abnormalities. Local injection site reactions were evaluated, and the diameters of all observed swellings were scored as small (<1.5 cm), medium (>1.5 to 5 cm), large (>5 to 10 cm), or were measured with a caliper. Blood samples were taken prior to vaccination, one day prior to challenge and 21 (*L. intracellularis* study), 28 (PCV2d and *M. hyopneumoniae* studies), 30 (*M. hyopneumoniae* DOI study) or 35 (PCV2a study) days after challenge. In the DOI studies, blood samples were also collected at regular intervals between vaccination and challenge. All challenge materials were of U.S. origin (Table 2). The pig breeds and number of males and females used in these studies are provided in Table 3. *M. hyopneumoniae* and *L. intracellularis* efficacy studies included three to five sentinel pigs per trial. The sentinel pigs were vaccinated with placebo and were necropsied prior to the challenge to confirm the absence of field infections and study validity.

#### 2.2.1. PCV2a Challenge Infection

For the PCV2a OOI study (performed October 2015 to January 2016), pigs were intranasally challenged at 7 weeks post vaccination (wpv) with a mixture of PCV2a and PRRSV given 3 mL per nostril at final titers of 2.5 and 7.2 log_10_TCID_50_ per pig, respectively. Both the PCV2a and PRRSV challenge viruses were isolated from tissues of diseased pigs supplied by Dr. Bruce Broderson at the University of Nebraska at Lincoln. Blood samples were taken weekly after challenge and analyzed to determine PCV2 viremia by quantitative polymerase chain reaction (qPCR) at the Iowa State University Veterinary Diagnostic Laboratory (ISU-VDL). Five weeks after challenge, all pigs were euthanized and a necropsy was performed to collect the tonsil, mesenteric lymph nodes and bronchial lymph nodes. The tissues were evaluated for lymphoid depletion after hematoxylin and eosin (H&E) staining at the ISU-VDL. The pathologist scored the tissues as 0 (normal), 1 (mild lymphoid depletion with loss of overall cellularity), 2 (moderate lymphoid depletion) or 3 (severe lymphoid depletion with loss of follicle structure). Scores > 1 were considered indicative of PCV2 systemic disease. For the PCV2 challenge titration, the challenge material was serially diluted, two-fold, and each dilution was added to 10 wells in each of two 96-well plates containing PK15 cells. The plates were incubated at 35–39 °C for five days, fixed with cold acetone and dried. Anti-PCV monoclonal antibody 3/1B4 was added to the wells, 50 μL/well, and the plates were incubated at 35–39 °C with shaking for one hour. The plates were washed, and then observed microscopically for fluorescence. For the PRRSV challenge titration, the challenge material was diluted serially, 10-fold, and each dilution was added to 10 wells in each of two 96-well plates containing MARC-145 cells. The plates were incubated in 5% CO_2_ at 35–39 °C for seven days. After the seven-day incubation, the plates were observed macroscopically for cytopathic effects. The wells were scored as positive or negative. For both titrations, the log_10_ TCID_50_/mL was calculated using the Spearman–Karber Method.

#### 2.2.2. PCV2d Challenge Infections

For the PCV2d studies (performed November 2019 to September 2021), pigs were intranasally challenged with virulent PCV2d at 4 wpv (OOI) or 16 wpv (DOI) with 3 mL per nostril at a final titer of 1.8 and 6.2 log_10_TCID_50_ per pig, respectively. The challenge virus was isolated from infected tissues procured from Dr. Richard Hesse at the Kansas State University Veterinary Diagnostic Laboratory (KSU-VDL). Blood samples were taken weekly after challenge and analyzed to determine PCV2 viremia by qPCR at the ISU-VDL. Five (OOI) or four (DOI) weeks after PCV2d challenge, all pigs were euthanized and a necropsy was performed to collect the tonsil, mesenteric lymph nodes, bronchial lymph nodes and Peyer’s Patches. The tissues were evaluated and scored for lymphoid depletion after H&E staining at the ISU-VDL, as described above. For the PCV2d challenge titration, the challenge material was serially diluted 5-fold from dilution 5^−1^ to 5^−8^. The dilutions were transferred to 96-well cell culture plates with 10 replicate wells per dilution. A suspension of PK-15 cells was then dispensed into all wells and the plates were incubated at 37 °C. After five days, the media were discarded, and the cells were fixed with 80% acetone. Plates were stained using an anti-PCV2 monoclonal antibody, followed by incubation with fluorescein-conjugated antimouse IgG. Positive wells were identified using a fluorescence microscope, and titers were calculated by the Spearman–Karber method. Titers were reported as log_10_ TCID_50_/mL.

#### 2.2.3. *M. Hyopneumoniae* Challenge Infection

The *M. hyopneumoniae* challenge was administered to pigs intratracheally at 5 wpv with 10 mL (OOI study performed May to August 2015), or at 10 wpv with 20 mL (DOI study performed March to September 2018) per pig containing 4.8 and 5.6 log_10_ Color Changing Units (CCU)/mL, respectively. The challenge inoculum was a lung homogenate of U.S. isolate strain 232 procured from the Iowa State University College of Veterinary Medicine. Twenty-eight to 30 days after challenge, necropsy was performed on all pigs, and the lungs were collected and evaluated for lung lesions. Lung lesions were scored by shading the approximate size and shape of the lesion onto a gridded outline of a lung and its associated lobes. The number of shaded squares was counted for the apical, cardiac, diaphragmatic, and intermediate lobes on both the dorsal and ventral sides. The total number of shaded squares was divided by the total number of squares in the lung grid representing total lung surface area, and then multiplied by 100 to calculate a percent lung lesion score (LLS). The final LLS was the average LLS of two blinded scorers.

#### 2.2.4. *L. Intracellularis* Challenge Infection

The *L. intracellularis* challenge was performed by orally inoculating pigs with 4.4 and 4.3 log_10_ TCID_50_ of an infected gut homogenate containing a U.S. isolate at 5 wpv (OOI study performed August to November 2019) or 20 wpv (DOI study performed April to December 2015), respectively. The challenge material was prepared as described previously (7). Briefly, mucosa from the guts of pigs with clinical signs of ileitis were collected by scraping, diluted in sucrose-phosphate-glutamate buffer, homogenized in a warren blender, trypsinized, aliquoted and frozen at −70 °C until use. Three weeks after the challenge, pigs were euthanized, and necropsy was performed to evaluate the intestine. The length of the affected area of the ileum was measured and the severity of the gross lesions was scored as 0 (normal mucosa), 1 (slight mucosal edema or slight hyperemia), 2 (moderate PPE, mucosal edema, hyperemia, some mucosal thickening), 3 (severe PPE, mucosal edema, severe hyperemia and mucosal corrugation) or 4 (severe PPE, score 3 plus hemorrhaging and/or necrosis, blood clots or yellowish pseudomembrane) by two scorers. Scores higher than 1 were considered indicative of ileitis related to the challenge. In addition, mucosa proximal to the ileo-cecal junction was collected for histopathological evaluation after hematoxylin and eosin (H&E) staining at the ISU-VDL. The pathologist scored the tissues as 0 (no diagnostic lesions), 1 (mild individual crypt proliferative change) or 2 (marked proliferative enterocolitis). Lastly, a mucosal sample from the ileo-cecal junction area was collected by scraping and assayed for bacterial load using qPCR. To confirm the *L. intracellularis* challenge dose, inocula were diluted 1:25 in antibiotic containing cell culture medium, followed by further two-fold dilutions. Dilutions of 1:25 to 1:800 were inoculated onto McCoy cells, grown to approximately 30% confluency in 96-well plates. Plates were incubated for six days at 37 °C in a humidified tri-gas incubator with 3% H_2_/97% N_2_ supply, set at 8% CO_2_, 8% O_2_. Following incubation, the inoculated monolayers were fixed and stained using a mouse anti-Lawsonia monoclonal antibody, followed by a fluorescein (FITC) labeled goat anti-mouse conjugate for reading on a fluorescence microscope. Titers were determined by the Spearman–Kärber method and were reported as the reciprocal of the highest dilution that gives a positive fluorescent signal. A subset of pigs was maintained until 7 weeks after the challenge and serological response was evaluated.

#### 2.2.5. Serology

To evaluate the serological response to vaccination, a PCV2 immunofluorescence assay (IFA) was performed at the KSU-VDL as previously described [9], and an *M. hyopneumoniae* commercial ELISA (IDEXX or DAKO M. hyo Ab test) was performed at the ISU-VDL as per the manufacturer’s directions.

For *L. intracellularis*, IFA was either performed in-house (DOI trial), or an immuno-peroxidase monolayer assay (IPMA) was performed at the University of Minnesota Veterinary Diagnostic Laboratory (UMN-VDL) as previously described (OOI trial) [10]. For the in-house IFA, serial dilutions of the sera were tested on microtiter plates containing fixed McCoy cells infected with *L. intracellularis*. After incubation with sera, the plates were rinsed and stained with a FITC-conjugated anti-swine IgG and titers were determined using a fluorescent microscope. The serum titer was calculated as the reciprocal of the highest serum dilution that gave a positive signal. For the IPMA, results were expressed as negative or positive according to standard procedures of the laboratory.

#### 2.2.6. Quantification of PCV2 DNA

Quantification of the PCV2 viral load in serum samples was performed by qPCR at the ISU-VDL as previously described [11]. All titers were reported as log_10_ DNA copies/mL.

#### 2.2.7. Quantification of Lawsonia DNA

Quantification of the Lawsonia bacterial load in ileo-cecal samples was performed by qPCR at the ISU-VDL as previously described [12] and was reported as threshold cycle number (Ct) or log_10_ DNA copies/mL. The Ct cut-off value was 40.

#### 2.2.8. Statistical Analysis

##### PCV2 Analysis

Analysis of the PCV2 viremia and lymphoid depletion of tonsil, mesenteric lymph node, bronchial lymph node, and Peyer’s Patch (PCV2d studies only) data was conducted using the R code PF (https://github.com/ABS-dev/PF, accessed on multiple dates from 2015 to 2019) package RRtosst module to perform the litter-coalesced prevented fraction analysis. Additionally, the litter-stratified prevented fraction (PF) analysis was conducted by calculating the confidence interval (CI) for the complement of the stratified risk ratio 1 − (Pv/Pc) from the Gart–Nam and/or Mantel–Haenszel estimators in the PF package in R-3.5.0. PCV2 serology data were analyzed by one-way ANOVA in MiniTab, v.20.

##### *M. Hyopneumoniae* Analysis

*M. hyopneumoniae* lung lesion scores of the vaccinate group were compared to the placebo group using the R code MFBoot module (https://github.com/ABS-dev/MF, accessed on multiple dates from 2015 to 2019) to estimate the litter-coalesced mitigated fraction (MF) and associated 95% confidence interval (CI) (highest density). The R code MFClusBoot module (https://github.com/ABS-dev/MF, accessed on multiple dates from 2015 to 2019) was used to estimate the litter-stratified (with resampling within the litter) MF (https://digitalcommons.wayne.edu/jmasm/vol4/iss2/14/, accessed on multiple dates from 2015 to 2019) and associated 95% CI (highest density) in R-3.5.0.

##### *L. Intracellularis* Analysis

Incidence of ileitis and colonization due to *L. intracellularis* were analyzed (where appropriate) using the litter-stratified prevented, or mitigated, fraction and the associated 95% CI, using the methods described above.

## 3. Results

### 3.1. Safety

Safety results of the OOI and DOI trials for each antigen fraction are summarized in Table 4.

In the laboratory studies, no animals developed systemic reactions. Injection site swellings were observed in 44% of the vaccinate and 40% of the placebo animals. The mean swelling size was 4.6 cm (maximum 11 cm) in the vaccinate and 4.1 cm (maximum 9.9 cm) for the placebo groups. The maximum duration was 14 consecutive days for the vaccinate and 13 consecutive days for the placebo groups.

### 3.2. Efficacy

A few vaccinated and control pigs had lameness, umbilical hernias or died due to blood collection. A total of four vaccinates (1%) and five controls (2%) were euthanized for welfare reasons.

#### 3.2.1. PCV2 Efficacy

PCV2 challenge did not result in clinical signs except for one pig in the placebo group, which had an occasional cough. Despite the lack of clinical signs, the viremia and lymphoid depletion data clearly demonstrated infection. Median virus DNA copies were 4.5 (PCV2a, OOI), 5.6 (PCV2d, OOI) and 2.4 (PCV2d, DOI) log_10_ lower in the vaccinated pigs at the time of peak viremia. The difference in the incidence of viremia between the groups was also significant for each trial. Lymphoid depletion scores in the vaccinate group were lower than the control pigs after challenge, and the differences between groups also met the statistical criteria for prevention. These data are found in Figure 1 (PCV2a, OOI, Appendix A), Figure 2 (PCV2d, OOI, Appendix A) and Figure 3 (PCV2d, DOI, Appendix A).

#### 3.2.2. *M. Hyopneumoniae* Efficacy

A serological response to vaccination was also seen in both *M. hyopneumoniae* challenge trials with 65% of vaccinated animals seropositive at both 5 wpv (Figure 4A; Appendix A) and 10 wpv (Figure 5A; Appendix A). Almost all the placebo animals responded serologically to the challenge infection. At necropsy, four weeks post-challenge, the median *M. hyopneumoniae*-induced lung lesions in the vaccinated groups were 50% (Figure 4B; OOI study, Appendix A) and 54% (Figure 5B; DOI study, Appendix A) lower than in the placebo groups. The differences in the lung score data between groups met the statistical criteria supporting the reduction in lung lesions caused by *M. hyopneumoniae* after vaccination with the tri-valent vaccine.

#### 3.2.3. *L. Intracellularis* Efficacy

Seroconversion after vaccination was demonstrated in both *L. intracellularis* challenge trials with 85% of the vaccinated animals seropositive at 5 wpv (Figure 6A; Appendix A) and 100% seropositive at 20 wpv (Figure 7A; Appendix A). Nearly all the placebo animals serologically responded to the challenge infection in the OOI study and all of the placebo animals serologically responded in the DOI study. At necropsy, three weeks post challenge, the median percentage of pigs with *L. intracellularis*-induced ileum gross lesion scores indicative of ileitis in the vaccinated groups were 74% (Figure 6B; OOI study, Appendix A) and 44% to 47% (Figure 7B; DOI study, Appendix A) lower than in the placebo groups, depending on the scorer. In addition, the incidence of *L. intracellularis*-induced histopathological lesions in the vaccinated groups was 48% (Figure 6C; OOI study, Appendix A) and 57% (Figure 7C; DOI study, Appendix A) lower than in the placebo groups. Finally, median bacterial loads at the ileo-cecal junction were 3.1 log_10_ (Figure 6D; OOI, Appendix A) or 6.4 Ct (Figure 7D; DOI, Appendix A) higher in the vaccinated pigs by qPCR. All data met the statistical criteria supporting the reduction in the incidence of ileitis caused by *L. intracellularis* after vaccination with the tri-valent vaccine.

## 4. Discussion

These studies demonstrate the safety and efficacy of a first tri-valent ready-to-use vaccine given once to pigs 3 weeks of age and older. In all vaccination trials, the vaccine was safe with moderate local reactions that were completely resolved on average, within 2 weeks. No systemic reactions were observed in these studies.

The laboratory challenge studies indicate that protection was observed as early as 4 (PCV2) and 5 weeks (*M. hyopneumoniae* and *L. intracellularis*) after vaccination and lasts for at least 10 weeks (*M. hyopneumoniae*), 16 weeks (PCV2d) and 20 weeks (*L. intracellularis*), respectively. Since there were no clinical signs in these controlled studies, the pathology that the tri-valent vaccine prevented includes PCV2 viremia and lymphoid depletion against both PCV2a (Figure 1B,C; Appendix A) and PCV2d (Figure 2B,C; Appendix A) infections in the OOI studies and PCV2d for the DOI study (Figure 3B,C; Appendix A). There was no correlation between serological response and protection, as demonstrated in the DOI study where only 43% of the vaccinated pigs were positive for PCV2 antibodies before challenge, but vaccination significantly protected against viremia and lymphoid depletion caused by a PCV2d infection (Figure 3B,C; Appendix A). This aligns with the notion that PCV2 protection is not only due to humoral immunity, but also cell mediated immunity, specifically memory T cells that secrete IFN-γ [13]. Currently, the most prevalent PCV2 genotype in the U.S. is PCV2d, with PCV2a still circulating in the population. In addition, PCV2 may coinfect with other pathogens, such as PCV3, thereby creating a high risk of porcine circovirus disease [13,14,15]. The tri-valent vaccine helps to reduce this risk by cross protecting against both genotype 2a and 2d and confirms previous observations by T. Kekarainen et al. and C-T Xiao et al. [15,16].

*M. hyopneumoniae* disease manifests as well-demarcated purple to plum-colored lung lesions without consolidation, usually in the apical, cardiac, intermediate and anterior diaphragmatic lobes and is associated with a non-productive cough in infected animals. *M. hyopneumoniae*-induced lung lesions were lower in both OOI (Figure 4B; Appendix A) and DOI studies (Figure 5B; Appendix A) by at least 50%. A reduction in this magnitude can have a significant effect on a herd because, like PCV2, *M. hyopneumoniae* is also a primary agent associated with PRDC. Vaccination against *M. hyopneumoniae* been shown to induce specific antibodies in both the serum and respiratory tract and it increases IFN-γ generation in the blood. This latter cell mediated response in vaccinated pigs is more noticeable than in non-vaccinated pigs. It has been reported that the use of a *M. hyopneumoniae* vaccine improves weight gain, feed conversion and at time mortality rate. Some studies have indicated that the use of vaccines may reduce the number of organisms in the respiratory tract and, as a result, decrease the infection level in a herd [17]. Further, the presence of *M. hyopneumoniae* has been shown to enhance PCV2 viremia in dually infected *M. hyopneumoniae*/PCV2 co-infected pigs [18]. *M. hyopneumoniae* disease manifests in non-vaccinated pigs at 16 to 20 weeks of age, whereas PCV2 disease is observed at one to 6 months of age [19,20]. Since the timing of *M. hyopneumoniae* vaccination with respect to PCV2 infection is essential, a tri-valent vaccine provides a solution as pigs are vaccinated with both *M. hyopneumoniae* and PCV2 at 3 weeks of age, which is well before the infection of either agent is anticipated in the field.

Pathology of *L. intracellularis* disease in pigs presents as thickening of the ileum and jejunum sections of the intestine due to the proliferation of immature enterocytes and near absence of inflammatory cells around infected crypts. The hemorrhagic form of infection occurs in older pigs and results in intestinal bleeding resulting in a high mortality. The intestines of these pigs are extended and thickened with the lumen filled with blood or fibrous blood clots. *L. intracellularis*-induced ileitis was greatly reduced after vaccination with a significant reduction in ileum lesions (74%) and histopathological lesions (58%) in the DOI study (Figure 7B,C; Appendix A). The OOI (Figure 6D; Appendix A) and DOI (Figure 7D; Appendix A) PCR results of ileum mucosal samples further supported the reduction in *L. intracellularis* colonization, as vaccinated animals had a 10- to 1000-fold reduction in the median amount of *L. intracellularis* in the ileo-cecal junction. This reduction implies that this tri-valent vaccine can reduce the shedding of *L. intracellularis* and, in turn, reduce the spread of the organism in a herd. Vaccination against *L. intracellularis* maintains the gut integrity and goblet cells resulting in a decrease in incidence, severity of disease and duration of bacterial shedding [21]. Disease caused by *L. intracellularis*, as well as *M. hyopneumoniae*, may result in a reduction in feed to gain ratio (F:G), as well as diminished lean accretion and average daily weight gain [20]. Together, these factors can have a significant impact on the bottom line for swine producers, including the health and welfare of their animals as well as their carbon footprint. Vaccination with a tri-valent, ready-to-use vaccine can ease the energy demand on resources allocated to growth, maintenance and immune function [22,23].

In conclusion, the trials successfully demonstrated the safety and efficacy of the first ready-to-use PCV2, *M. hyopneumoniae*, and *L. intracellularis* vaccine.

## 5. Conclusions

Under the conditions of this study, the following conclusions were reached:The trials successfully demonstrated the safety and efficacy of the first ready-to-use PCV2, *M. hyopneumoniae*, and *L. intracellularis* vaccine.The use of a tri-valent, ready to use vaccine, containing Porcine Circovirus Type 2a, *M. hyopneumoniae*, and *L. intracellularis* results in fewer injections, reducing animal stress.The use of the tri-valent vaccine supports a smaller carbon-footprint by reducing inventory and waste.The use of the tri-valent vaccine reduces labor needs during vaccination.

## Figures and Tables

**Figure 1 vaccines-13-00681-f001:**
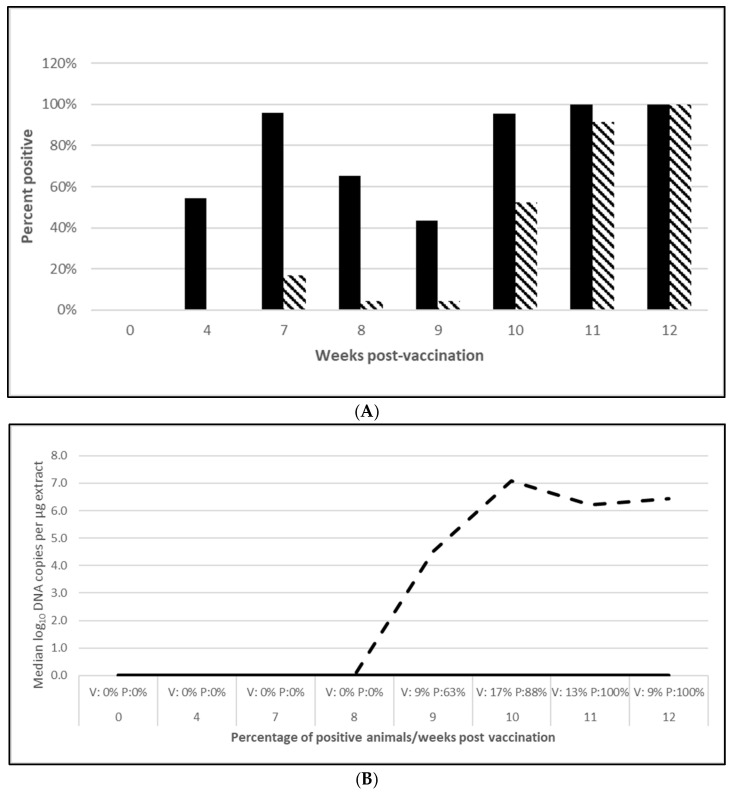
Porcine circovirus type 2a onset of immunity laboratory study data. All pigs were given virulent PCV type 2a (PCV2a) and porcine reproductive and respiratory syndrome virus (PRRSV) 7 weeks post vaccination (wpv). (**A**) is the percentage of animals with a positive anti-PCV2 response. The solid bar represents the vaccine group and the patterned bar the placebo. All pigs in the vaccinate group had significantly higher antibody titers at 4 wpv ( *p* = 0.00), 7 wpv (*p* = 0.00), 8 wpv (*p* = 0.001), and 9 wpv (*p* = 0.007). At 12 wpv, 5 weeks post challenge (wpc), the placebo group had significantly higher antibody titers (*p* = 0.002). The data were analyzed by ANOVA. (**B**) is the median PCV DNA load in serum samples including the percentage of PCV2-positive animals. In the vaccine group, the median was zero for all tissues. The dashed lines represents the placebo group. Below the x-axis, V denotes the percentage of vaccinated pigs that were positive for viremia whereas P is the placebo group. The vaccinate group had significantly lower viremia compared to the placebo group (prevented fraction (PF) of 0.79 and a 95% confidence interval (CI) of 0.60 to 0.91). (**C**) is the percentage of pigs with a lymphoid depletion score greater than 1, including the median lymphoid depletion score. All of the vaccinated pigs were negative for lymphoid depletion. The patterned bar represents the placebo group. Below the x-axis, V denotes the percentage of vaccinated pigs that were positive for viremia whereas P is the placebo group. Lymphoid depletion was significantly lower in the vaccinate group compared with the placebo group with a PF of 1.0. The data for panels (**B**,**C**) were analyzed by calculating the CI for the complement of the stratified risk ratio 1 − (Pv/Pc) from the Gart–Nam and/or Mantel–Haenszel estimators in the PF package in R-3.5.0.

**Figure 2 vaccines-13-00681-f002:**
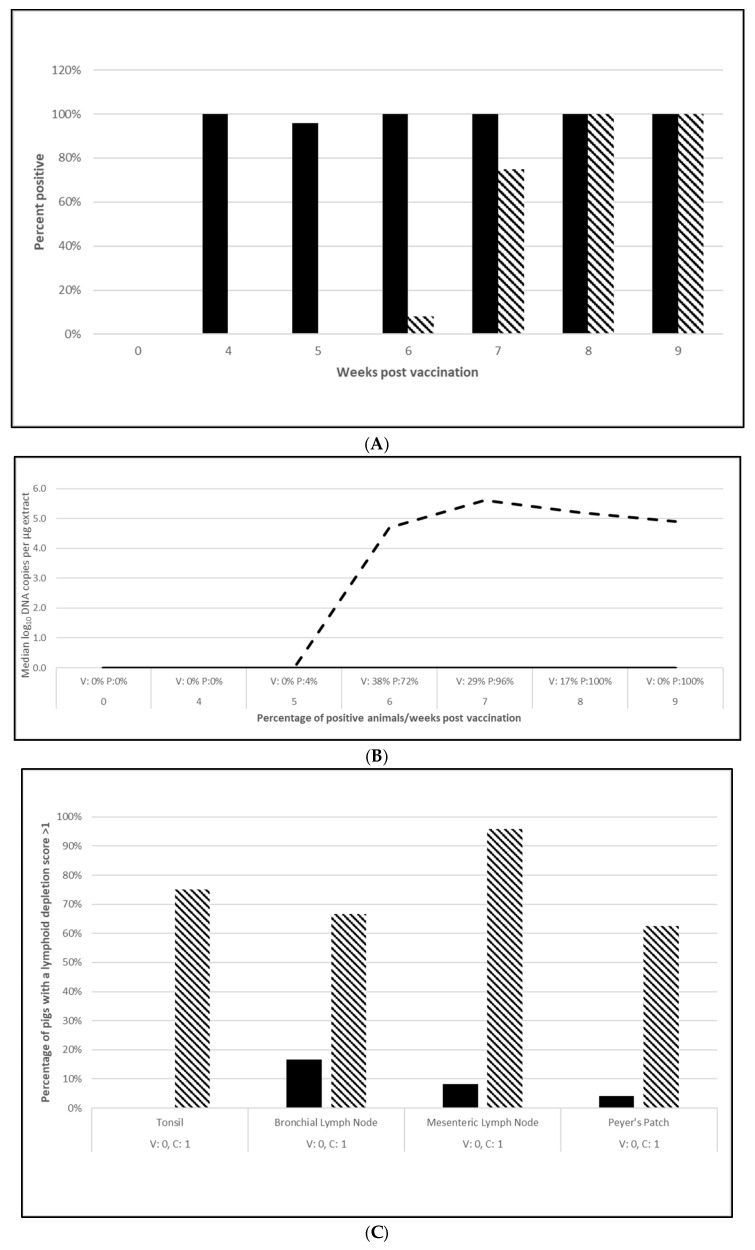
Porcine circovirus type 2d onset of immunity laboratory study data. All pigs were given virulent PCV type 2d (PCV2d) 4 wpv. (**A**) is the percentage of animals with a positive anti-PCV2 response. The solid bar represents the vaccine group and the patterned bar the placebo. All pigs in the vaccinate group had significantly higher antibody titers at 4 wpv (*p* = 0.00), 5 wpv (*p* = 0.00), 6 wpv (*p* = 0.00), and 7 wpv (*p* = 0.023). At 4 and 5 wpc (8 and 9 wpv), the placebo group had significantly higher antibody titers (*p* = 0.00 for both). The data were analyzed by ANOVA. (**B**) is the median PCV DNA load in serum samples including the percentage of PCV2-positive animals. The solid line represents the vaccine group and the dashed line the placebo. Below the x-axis, V denotes the percentage of vaccinated pigs that were positive for viremia whereas P is the placebo group. The vaccinate group had significantly lower viremia compared to the placebo group (PF of 0.45 and a 95% CI of 0.17 to 0.64). (**C**) is the median percentage of pigs with a lymphoid depletion score greater than 1, including the median lymphoid depletion score. The solid bar represents the vaccine group and the patterned bar the placebo. Below the x-axis, V denotes the percentage of vaccinated pigs that were positive for lymphoid depletion whereas P is the placebo group. Lymphoid depletion was significantly lower in the vaccinate group as compared with the placebo group (PF of 0.76 and a 95% CI of 0.51 to 0.89). The data for panels (**B**,**C**) were analyzed by calculating the CI for the complement of the stratified risk ratio 1 − (Pv/Pc) from the Gart–Nam and/or Mantel–Haenszel estimators in the PF package in R-3.5.0.

**Figure 3 vaccines-13-00681-f003:**
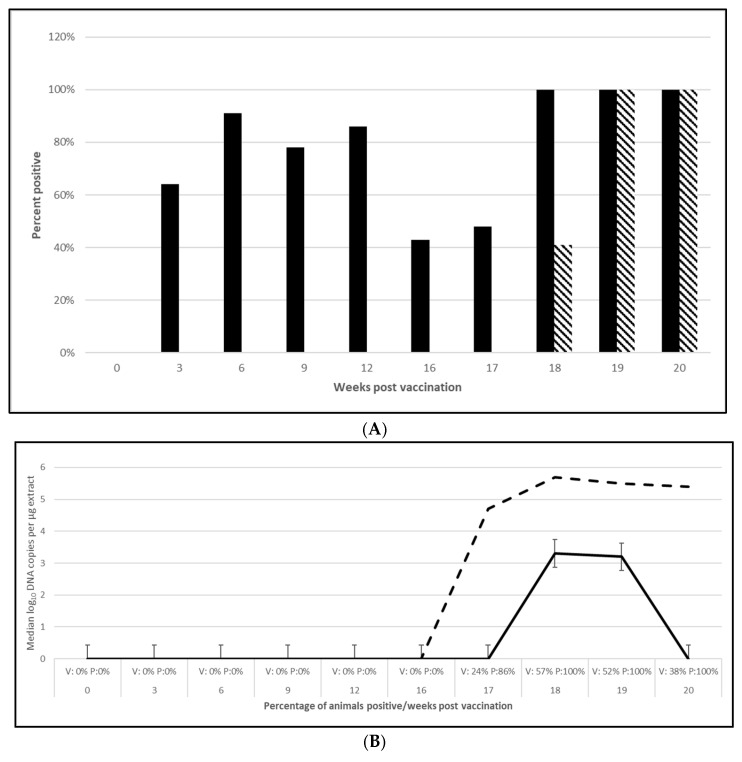
Porcine circovirus type 2d duration of immunity laboratory study data. All pigs were given virulent PCV2d 16 wpv. (**A**) is the percentage of animals with a positive anti-PCV2 response. The solid bar represents the vaccine group and the patterned bar the placebo. All pigs in the vaccinate group had significantly higher antibody titers at 3, 6, 9, 12, and 16 wpv (*p* = 0.00 for all). The vaccine group also had significantly higher antibody titers at 1 and 2 wpc (17 and 18 wpv), (*p* = 0.024 and 0.00, respectively), whereas the placebo group had higher antibody titers than the vaccinates at 4 wpv, or 20 wpv, (*p* = 0.00). The data were analyzed by ANOVA. (**B**) is the median PCV DNA load in serum samples including the percentage of PCV2-positive animals. The solid line represents the vaccine group and the dashed line the placebo. Below the x-axis, V denotes the percentage of vaccinated pigs that were positive for viremia, whereas P is the placebo group. The vaccinate group had significantly lower viremia compared to the placebo group (PF of 0.33 and a 95% CI of 0.11 to 0.49). (**C**) is the median percentage of pigs with a lymphoid depletion score greater than 1, including the median lymphoid depletion score. The solid bar represents the vaccine group and the patterned bar the placebo. Below the x-axis, V denotes the percentage of vaccinated pigs that were positive for lymphoid depletion whereas P is the placebo group. Lymphoid depletion was significantly lower in the vaccinate group compared with the placebo group (PF of 0.46 and a 95% CI of 0.17 to 0.65). The data for panels (**B**,**C**) were analyzed by calculating the CI for the complement of the stratified risk ratio 1 − (Pv/Pc) from the Gart–Nam and/or Mantel–Haenszel estimators in the PF package in R-3.5.0.

**Figure 4 vaccines-13-00681-f004:**
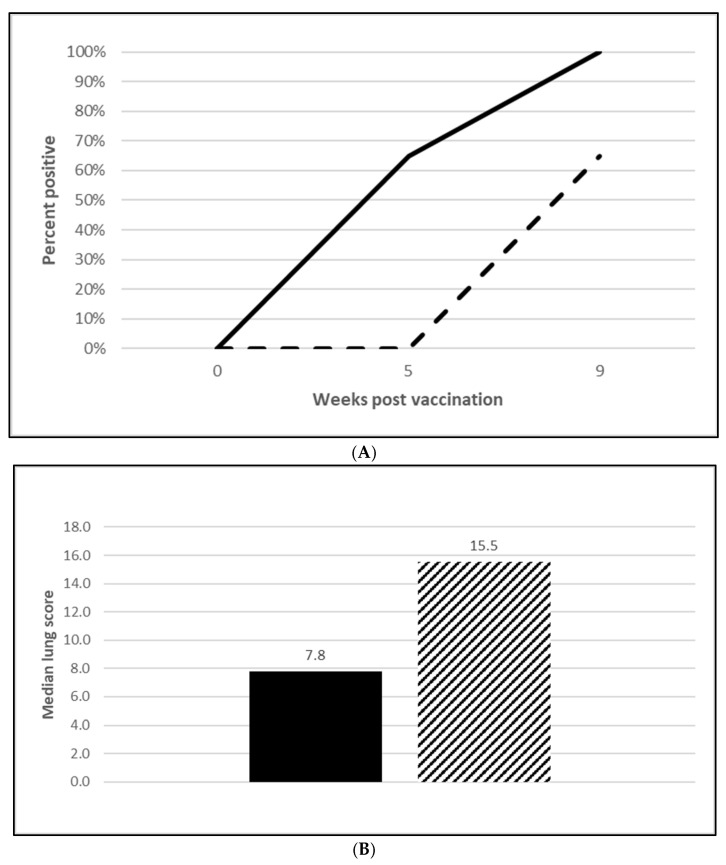
*Mycoplasma hyopneumoniae* onset of immunity laboratory study data. All pigs were given virulent *M. hyopneumoniae* 5 wpv. (**A**) is the percentage of animals with a positive anti- *M. hyopneumoniae* response. The solid line represents the vaccine group and the dashed line the placebo. (**B**) is the median post challenge lung score data. The solid bar represents the vaccine group and the patterned bar the placebo. The vaccinate group had significantly lower lung scores compared to the placebo group (mitigated fraction (MF) of 0.72 and a 95% CI of 0.42 to 1.0). The data were analyzed using the R code MFClusBoot module to estimate the litter-stratified (with resampling within the litter).

**Figure 5 vaccines-13-00681-f005:**
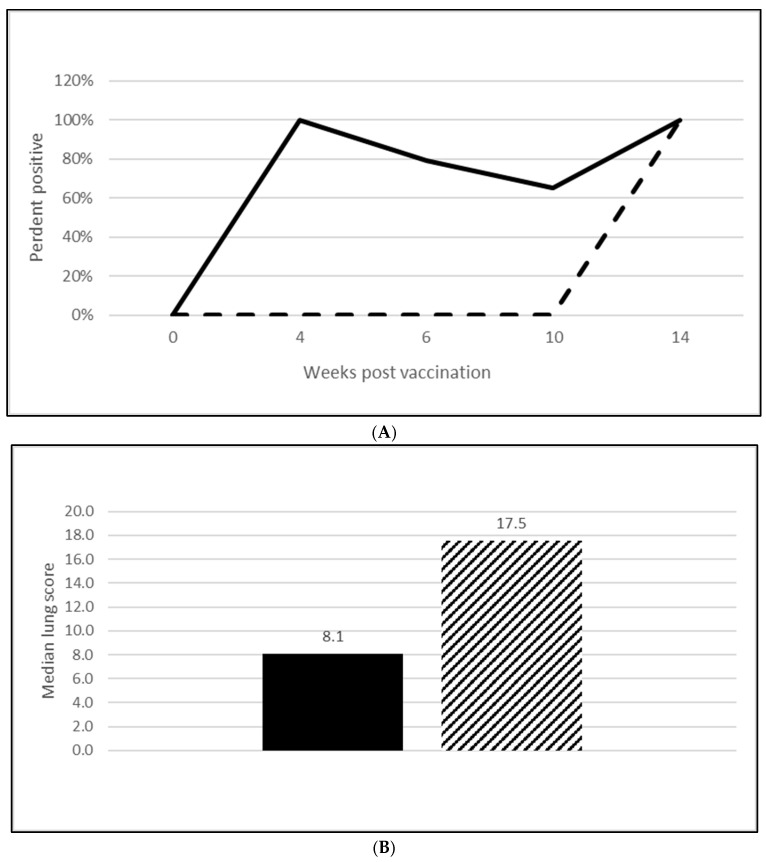
*Mycoplasma hyopneumoniae* duration of immunity laboratory study data. All pigs were given virulent *M. hyopneumoniae* 10 wpv. (**A**) is the percentage of animals with a positive anti-*M. hyopneumoniae* response. The solid line represents the vaccine group and the dashed line the placebo. (**B**) is the median post challenge lung score data. The solid bar represents the vaccine group and the patterned bar the placebo. The vaccinate group had significantly lower lung scores compared to the placebo group (MF of 0.57 and a 95% CI of 0.33 to 0.78). The data were analyzed using the R code MFClusBoot module to estimate the litter-stratified (with resampling within the litter) MF3 and associated 95% CI (highest density) in R-3.5.0.

**Figure 6 vaccines-13-00681-f006:**
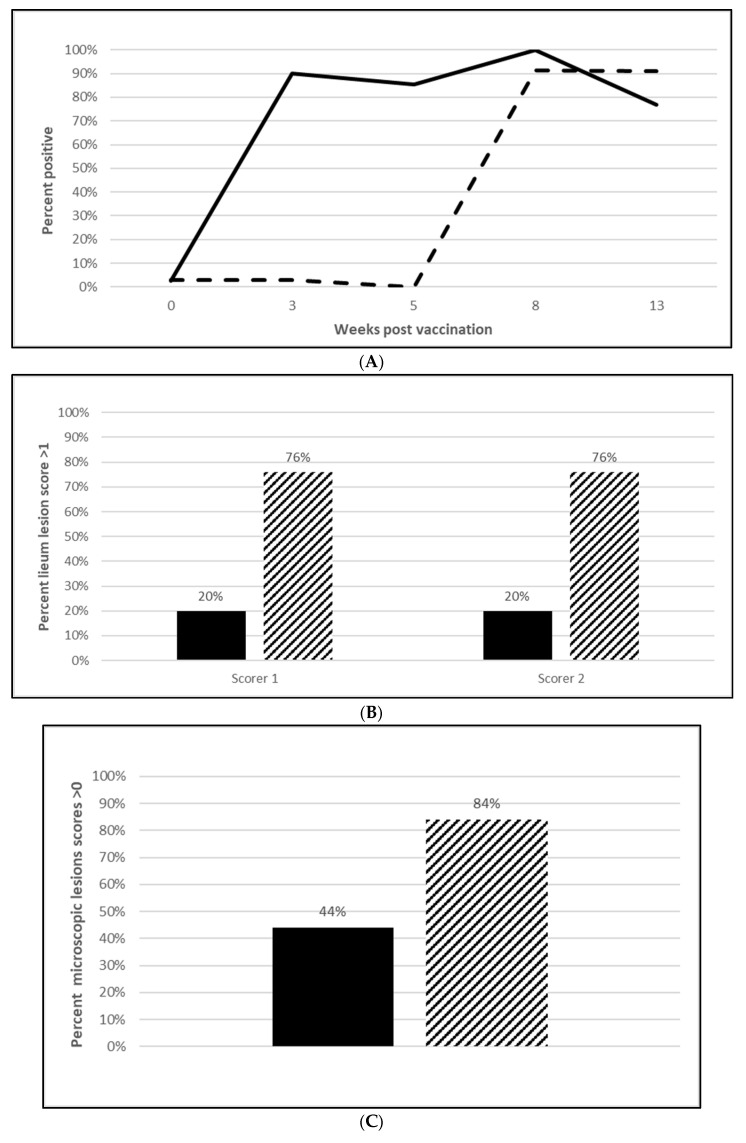
*Lawsonia intracellularis* onset of immunity laboratory study data. All pigs were given virulent *L. intracellularis* 5 wpv. (**A**) is the percentage of animals with a positive anti-*L. intracellularis* response. The solid line represents the vaccine group and the dotted line the placebo. (**B**) is the percentage of animals with ileum scores > 1. The solid bar represents the vaccine group and the patterned bar the placebo. The vaccinate group had significantly lower ileum scores compared to the placebo group (PF of 0.71 and a 95% CI of 0.40 to 0.86). (**C**) is the percentage of pigs with a histopathological lesion score > 0. The solid bar represents the vaccine group and the patterned bar the placebo. Histopathological lesion scores were significantly lower in the vaccinate group as compared with the placebo group (PF of 0.51 and a 95% CI of 0.23 to 0.68). The data for panels (**B**,**C**) were analyzed by calculating the CI for the complement of the stratified risk ratio 1 − (Pv/Pc) from the Gart–Nam and/or Mantel–Haenszel estimators in the PF package in R-3.5.0. (**D**) is the bacterial load of *L. intracellularis*, in log_10_ DNA of copies, per mL of ilio-cecal samples. The solid bar represents the vaccine group and the patterned bar the placebo. The vaccinate group had significantly lower *L. intracellularis* DNA at the ilio-cecal junction compared to the placebo group (MF of 0.69 and a 95% CI of 0.38 to 1.0). The data were analyzed using the R code MFClusBoot module to estimate the litter-stratified (with resampling within the litter) MF and associated 95% CI (highest density) in R-3.5.0. The data were analyzed using the R code MFClusBoot module to estimate the litter-stratified (with resampling within the litter) MF and associated 95% CI (highest density) in R-3.5.0.

**Figure 7 vaccines-13-00681-f007:**
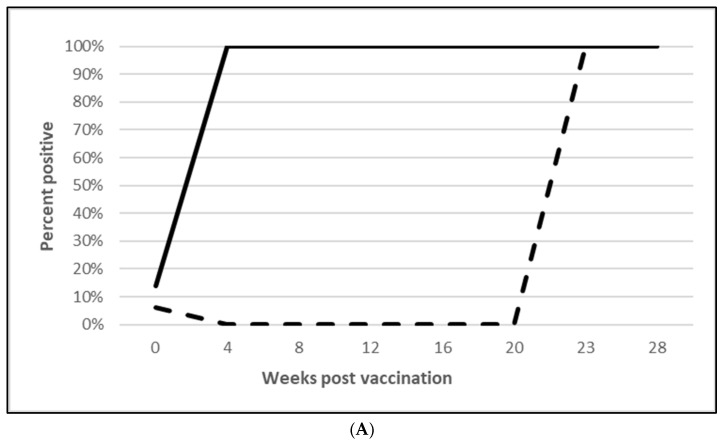
*Lawsonia intracellularis* duration of immunity laboratory study data. All pigs were given virulent *L. intracellularis* 20 wpv. (**A**) is the percentage of animals with a positive anti-*L. intracellularis* response. The solid line represents the vaccine group and the dotted line the placebo. (**B**) is the percentage of animals with ileum scores > 1. The solid bar represents the vaccine group and the patterned bar the placebo. The vaccinate group had significantly lower ileum scores compared to the placebo group (PF of 0.50 and a 95% CI of 0.32 to 0.68). (**C**) is the percentage of pigs with a histopathological lesion score > 0. The solid bar represents the vaccine group and the patterned bar the placebo. Histopathological lesion scores were significantly lower in the vaccinate group as compared with the placebo group (PF of 0.59 and a 95% CI of 0.41 to 0.75). The data for panels (**B**,**C**) were analyzed by calculating the CI for the complement of the stratified risk ratio 1 − (Pv/Pc) from the Gart–Nam and/or Mantel–Haenszel estimators in the PF package in R-3.5.0. (**D**) is the bacterial load of *L. intracellularis*, Cts, per mL of ilio-cecal samples. The solid bar represents the vaccine group and the patterned bar the placebo. The vaccinate group had significantly lower *L. intracellularis* DNA at the ilio-cecal junction compared to the placebo group (MF of 0.56 and a 95% CI of 0.25 to 0.81). The data were analyzed using the R code MFClusBoot module to estimate the litter-stratified (with resampling within the litter) MF and associated 95% CI (highest density) in R-3.5.0. The data were analyzed using the R code MFClusBoot module to estimate the litter-stratified (with resampling within the litter) MF and associated 95% CI (highest density) in R-3.5.0.

**Table 1 vaccines-13-00681-t001:** Overview of laboratory challenge studies. All piglets were vaccinated at three weeks of age or older.

Type of Study	Group	No. of Piglets	Challenge (Weeks Post-Vaccination)	Challenge Organism
Onset of Immunity	Circumvent CML	25	7	PCV2a/PRRSv *
	Placebo	25		
Onset of Immunity	Circumvent CML	25	4	PCV2d †
	Placebo	25		
Duration of Immunity	Circumvent CML	25	16	PCV2d
	Placebo	25		
Onset of Immunity	Circumvent CML	35	5	*Mycoplasma hyopneumoniae*
	Placebo	35		
Duration of Immunity	Circumvent CML	70	10	*Mycoplasma hyopneumoniae*
	Placebo	70		
Onset of Immunity	Circumvent CML	40	5	*Lawsonia intracellularis*
	Placebo	40		
Duration of Immunity	Circumvent CML	50	20	*Lawsonia intracellularis*
	Placebo	50		

* Porcine circovirus type 2a/porcine reproductive and respiratory syndrome virus. † Porcine circovirus type 2d.

**Table 2 vaccines-13-00681-t002:** Diagram of laboratory challenge studies.

	Weeks																		
	SD * 0	SD 3	SD 4	SD 5	SD 6	SD 7	SD 8	SD 9	SD 10	SD 12	SD 13	SD 14	SD 16	SD 17	SD 18	SD 19	SD 20	SD 23	SD 27
**PCV2a OOI ****	Vaccination with Circumvent CML or Placebo, Blood Sample		Blood Sample			PCV2a/PRRSV Challenge Blood Sample	Blood Sample (Weekly)			Collect Lymph Nodes, Blood Sample									
**PCV2d OOI**			PCV2d Challenge Blood Sample	Blood Sample (Weekly)				Collect Lymph Nodes, Blood Sample											
**PCV2d DOI *****		Blood Sample			Blood Sample			Blood Sample		Blood Sample			PCV2d Challenge Blood Sample	Blood Sample (Weekly)			Collect Lymph Nodes Peyer’s Patches Blood Sample		
***M. hyopneumoniae* OOI**				*M. hyopneumoniae* Challenge, Blood Sample				Score Lung Lesions Blood Sample											
***M. hyopneumoniae* DOI**			Blood Sample		Blood Sample				*M. hyopneumoniae* Challenge, Blood Sample			Score Lung Lesions Blood Sample							
***L. intracellularis* OOI**		Blood Sample		*L. intracellularis* Challenge, Blood Sample			Collect/Score Ileum Blood Sample				Blood Sample								
***L. intracellularis* DOI**			Blood Sample				Blood Sample			Blood Sample			Blood Sample				*L. intracellularis* Challenge, Blood Sample	Collect/Score Ileum Blood Sample	Blood Sample

* SD = Study Day in Weeks. ** OOI = Onset of Immunity. *** DOI = Duration of Immunity.

**Table 3 vaccines-13-00681-t003:** Description of pigs used in efficacy studies.

	Number of Pigs		Breed
	Male	Female	
PCV2a OOI *	24	25	Large White/Duroc
PCV2d OOI	33	17	Yorkshire/Landrace
PCV2d DOI **	32	18	Large White/Yorkshire
*M. hyopneumoniae* OOI	32	38	Yorkshire/Landrace
*M. hyopneumoniae* DOI ***	36	32	Yorkshire/Landrace
	39	31	Large White/Landrace
*L. intracellularis* OOI	44	36	Yorkshire/Landrace/Duroc
*L. intracellularis* DOI	48	52	Yorkshire/Landrace/Duroc

* OOI = Onset of Immunity. ** DOI = Duration of Immunity. *** Two pigs are not included in the pig count due to mislabeling.

**Table 4 vaccines-13-00681-t004:** Safety of the CML combination vaccine.

Study	Details	Vaccine	Placebo
PCV2a OOI	Number of pigs (n)	25	24
	Pigs with local reactions (%)	88	88
	Pigs with a systemic reaction (%)	0	0
PCV2d OOI	Number of pigs (n)	25	25
	Pigs with local reactions (%)	48	52
	Pigs with a systemic reaction (%)	0	0
PCV2d DOI	Number of pigs (n)	25	25
	Pigs with local reactions (%)	0	0
	Pigs with a systemic reaction (%)	0	0
*M. hyopneumoniae* OOI	Number of pigs (n)	35	35
	Pigs with local reactions (%)	3	9
	Pigs with a systemic reaction (%)	0	0
*M. hyopneumoniae* DOI	Number of pigs (n)	70	80
	Pigs with local reactions (%)	44	54
	Pigs with a systemic reaction (%)	0	0
*L. intracellularis* OOI	Number of pigs (n)	40	44
	Pigs with local reactions (%)	55	36
	Pigs with a systemic reaction (%)	0	0
*L. intracellularis* DOI	Number of pigs (n)	50	50
	Pigs with local reactions (%)	68	42
	Pigs with a systemic reaction (%)	0	0
All studies	Number of pigs (n)	270	283
	Pigs with local reactions (%)	44	40

## Data Availability

The data presented in this study are available on request from the corresponding authors.

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
