# Peer review of "Efficacy and Safety of a Tri-Valent Ready-to-Use Porcine Circovirus Type 2a, Mycoplasma hyopneumoniae and Lawsonia intracellularis Vaccine in Weaned Pigs"

_vaccines, 2025, doi:10.3390/vaccines13070681_

Round 1

Reviewer 1 Report

Comments and Suggestions for Authors

The manuscript by Allen et al. investigated the efficacy and safety of a tri-valent ready-to-use vaccine against PCV2a, PCV2d, Mycoplasma hyopneumoniae, and Lawsonia intracellularis in weaned pigs. Vaccine used in this study is a combination of baculovirus-expressed ORF2 capsid protein of PCV2, inactivated M. hyopneumoniae, inactivated L. intracellularis, and the XSOLVE adjuvant. However, all information on the pathogens used for vaccine manufacturing and challenge infection is absent, including strain names and NCBI accession numbers. Furthermore, how much PCV ORF2 capsid protein, inactivated Mycoplasma hyopneumoniae, and inactivated Lawsonia intracellularis are present in a single dose? This information should be explicitly stated in the Materials and Methods section. This manuscript lacks a thorough pathology analysis, which would provide more detailed information on the effects of the vaccine on the tissues and organs of the animals.

Specific comments:

  1. In the PCV2a challenge research, why did you challenge a combination of PCV2a and PRRSV? The challenge models utilized in the study may not accurately represent actual infection dynamics cause of the co-infection with PRRSV.
  2. This study provides limited discussion of the mechanisms by which the vaccine protects against the different pathogens. It would be helpful to include more information on the immune responses elicited by the vaccine and how these responses contribute to protection.
  3. Suggest adding a diagram of the animal experimental design.
  4. The results are presented as group means, which may obscure individual animal variation. It would be helpful to include individual animal data or summary statistics such as standard deviations.
  5. The study was funded by Intervet Inc. (d/b/a Merck Animal Health), and all authors are employees of the company, which could introduce potential bias.

Reviewer 2 Report

Comments and Suggestions for Authors

Authors have performed a cohort study with purpose of testing efficacy of vaccine afgainst three pathogens.

  1. Plaese provide more information on pigs used in the study – cohort study uses uniform groups, so information on breed, weight and sex would be beneficial.
  2. If it is written that challenge time is not applicable for sentinel animals, why are they in the Table 1?
  3. Include how many animals developed clinical disease.
  4. Did you monitor clinical signs - appetite, excretions, respiration, gait, body surface appearance...?
  5. Provide information on censored animals – in which challenge they participated and measures used to reduce bias in further calculations
  6. Would be interesting to see how vaccine performs in the case of PCV2 and PCV3 coinfection
  7. Line 407 – delete repeated words
  8. Line 504 – full name of the Journal

Reviewer 3 Report

Comments and Suggestions for Authors

In this manuscript, the authors described multiple trials demonstrating the safetyand efficacy of a tri-valent vaccine against diseases caused by PCV2a and PCV2d, Mycoplasma hyopneumoniae, and Lawsonia intracellularis. The results indicated that this tri-valent vaccine could reduce viremia and lymphoid depletion due to PCV2a and PCV2d, reduced lung lesions due to M. hyopneumonine and reduced ileumand colonization scores due to L. intracellularis. The results are positive, but many raw data, especially pathological detection data, were not provided by the authors. Below are the specific comments.

  1. 2.1. Vaccines, “inactivated M. hyopneumoniae, inactivated L. intracellularis”, please provide a detailed explanation of the generations of these two pathogens, how they were cultured, how they were inactivated, and how vaccines were formulated.
  2. 2.2.4. L. intracellularis Challenge Infection, “orally inoculating pigs with 4.4 and 4.3 log10 TCID50 of a U.S isolate”, please explain is the L. intracellularis used for challenge prepared from tissue homogenization? Or prepared from a cell culture strain? How was the calculation of TCID50 of L. intracellularis?
  3. Please provide the original histopathological images and corresponding scores for each piglet of the following results: (1) Figure 1C, Figure 2C and Figure 3C, lymphoid depletion score; (2) Figure 4B and Figure 5B, lung score. (3) Figure 6B and Figure 7B, ileum scores; (4) Figure 6C and Figure 7C, histopathological lesion scores.

Round 2

Reviewer 1 Report

Comments and Suggestions for Authors

Thank you for the responses. Pathology analysis is crucial in the context of vaccine development. I agree to extend the revision submission deadline to allow authors to complete pathology analysis and even explore further the mechanisms by which the vaccine protects against the different pathogens, as authors requested.

Author Response

See enclosure

Reviewer 3 Report

Comments and Suggestions for Authors

(1) In section 2.1., the authors introduced a commercialized vaccine (CIRCUMVENT CML, Intervet, Inc.), and then evaluated the safety and efficacy of this vaccine on three-week-old pigs. What is controversial is: If this is a commercialized vaccine that has obtained a veterinary drug production certificate, there is no need to conduct further research in this paper. In other words, these research contents should be completed in advance before the vaccine obtains a veterinary drug production certificate.

Are researchers conducting the safety and efficacy studies on their company's vaccine products due to insufficient experimental data during the vaccine registration period?

A commercialized vaccine will not mention specific generational information of the strains used for vaccine preparation in its instructions. As a scientific research, it is necessary to provide readers with this information in order to better understand and supervise the scientificity and rationality of this study.

(2) The authors did not clarify how TCID50 was calculated, and detailed data from relevant experiments, such as cytopathic effect (CPE) images or indirect immunofluorescence assay (IFA) images, need to be provided

(3) The original histopathological images of (1) Figure 1C, Figure 2C and Figure 3C, lymphoid depletion score; (2) Figure 4B and Figure 5B, lung score. (3) Figure 6B and Figure 7B, ileum scores; (4) Figure 6C and Figure 7C, are the core data of this study, unfortunately, the authors are unable to provide these images.

(4) All the data did not show variance, did the author only conduct one relevant experiment?

Author Response

See enclosure

Round 3

Reviewer 1 Report

Comments and Suggestions for Authors

No further comments.

Author Response

Reviewer 1 mentions 'No further comments'. Therefore, no additional information is presented by the authors.

Reviewer 3 Report

Comments and Suggestions for Authors

(1) Line 168-169, “---orally inoculating pigs with 4.4 and 4.3 log10 TCID50 of an infected gut homogenate containing ---”.

Did the author use SPF pigs to prepare the gut homogenate? Please provide a detailed preparation method.

How to ensure that the original L. intracellularis (gut homogenate?) and the later prepared gut homogenate used for challenge do not contain other pathogens that can cause disease in pigs, including certain immunosuppressive pathogens that can enhance the incidence of other pathogens? How did the author conduct the testing? Please provide the testing methods and results.

(2) 2.2.5. Serology, please provide original images of all IFA or IPMA test results in the corresponding results section. These original images are the key data that led to the conclusion of this manuscript.

(3) Similarly, the authors should provide all original images of gross lesions and H&E staining, rather than just providing scoring data.

Author Response

  1. Summary

Thank you very much for taking the time to review this manuscript. Please find the detailed responses below and the corresponding revisions/corrections highlighted/in track changes in the re-submitted manuscript.

  1. Point-by-point response to Comments and Suggestions for Authors

Comment (1):

Line 168-169, “---orally inoculating pigs with 4.4 and 4.3 log10 TCID50 of an infected gut homogenate containing ---”.

  • Did the author use SPF pigs to prepare the gut homogenate? Please provide a detailed preparation method.

Response: Thank you for your question. The method we used to prepare the gut homogenate was previously described in lit.ref. 7. We have updated the manuscript to include this reference and a brief description of the method (see line 175-178).

We endeavored to challenge the study pigs with a preparation which is representative of a natural infection. SPF pigs, while not harboring certain pathogens, nonetheless contain the normal intestinal flora. Therefore, gut homogenate from SPF pigs will still not be entirely free of other live organisms. We have previously tried to challenge pigs with tissue culture derived L. intracellularis, but found that the pathogenicity of the challenge material was greatly reduced and the challenge did not result in significant intestinal lesions (both gross and histopathological). For this reason we chose to conduct the reported studies for L. intracellularis with gut homogenate from experimentally challenged animals.

  • How to ensure that the original L. intracellularis (gut homogenate?) and the later prepared gut homogenate used for challenge do not contain other pathogens that can cause disease in pigs, including certain immunosuppressive pathogens that can enhance the incidence of other pathogens? How did the author conduct the testing? Please provide the testing methods and results.

Response: Thank you for your thoughtful comment. It is true that tissue cultured challenge material is usually prefered as this material can be defined better. However as mentioned above, we have found that passage of L. intracellularis in tissue culture results in significant attenuation of the bacteria. Therefore use of tissue cultured L. intracellularis was not appropriate for the purpose of the study, namely to replicate ileitis caused by L. intracellularis in conventionally raised pigs.

We used a carefully structured study design to assure that unintended infections could be prevented and identified. The source herd of the study animals was high-health, with a routine monitoring program for the health status of the pigs. Pigs were pre-screened prior to study selection for selected pathogens typically relevant for pig production such as PRRSv, SIV and the study pathogens. The studies were conducted indoors in controlled environments. Further, approximately equal numbers of pigs were enrolled in vaccine and control groups, and the groups were co-mingled within pens. For all trials, after challenge the control pigs allowed us to monitor for the appearance of any pathogenicity not attributable to the challenge infection. Also, for all M. hyopneumoniae and L. intracellularis studies, sentinel pigs were maintained during the vaccination phase of the studies. The sentinels were removed prior to the challenge and served to confirm that absence of opportunistic infections prior to the challenge. The use of sentinel pigs during the studies was not previously included in the manuscript but was now added in lines 93-96.

Comment (2):

2.2.5. Serology, please provide original images of all IFA or IPMA test results in the corresponding results section. These original images are the key data that led to the conclusion of this manuscript.

Response: Thank you for this comment. The serological testing was conducted at accredited Veterinary Diagnostic Laboratories in the US, according to applicable routine methods. The labs were located at Kansas State University (PCV testing), Iowa State University (M. hyopneumoniae testing) and University of Minnesota (L. intracellularis testing). The use of different laboratories for different pathogens is due to availability of certain test methods at certain D-labs. We find, for instance, that the IPMA test used for L. intracellularis testing is only available at the University of Minnesota. Since the test methods and resulting findings (cell pathology) were routine and unremarkable, no images were captured by the D-labs.   

Comment (3):

Similarly, the authors should provide all original images of gross lesions and H&E staining, rather than just providing scoring data.

Response: Thank you for the comment. We do not have original histopathological images available for the listed Figures. Lesions of PCV, M. hyopneumoniae and L. intracellularis have been described and images are commonly available in the literature and standard text books. Therefore images were not collected at the time. The histopathological evaluation of the tissues for these trials was conducted by pathologists at the Iowa State University – Veterinary Diagnostic Laboratory (ISU-VDL) according to standard practices. The ISU-VDL is an accredited diagnostic laboratory in the US. Also, the specifics of the scoring systems are included in the manuscript so as to allow the reader to understand the pathology that was observed (for L. intracellularis, for instance, see lines 181-184 for gross lesion scoring and lines 188-189 for histopathology due to L. intracellularis).

We have clarified the involvement of D-Lab pathologists in the scoring of the L. intracellularis studies in lines 187-188.